# Recurrent turnover of senescent cells during regeneration of a complex structure

Maximina H Yun[1]*, Hongorzul Davaapil[2], Jeremy P Brockes[1]

[1]Institute of Structural and Molecular Biology, Division of Biosciences, University College London, London, United Kingdom; [2]Institute of Ophthalmology, University College London, London, United Kingdom

**Abstract** Cellular senescence has been recently linked to the promotion of age-related pathologies, including a decline in regenerative capacity. While such capacity deteriorates with age in mammals, it remains intact in species such as salamanders, which have an extensive repertoire of regeneration and can undergo multiple episodes through their lifespan. Here we show that, surprisingly, there is a significant induction of cellular senescence during salamander limb regeneration, but that rapid and effective mechanisms of senescent cell clearance operate in normal and regenerating tissues. Furthermore, the number of senescent cells does not increase upon repetitive amputation or ageing, in contrast to mammals. Finally, we identify the macrophage as a critical player in this efficient senescent cell clearance mechanism. We propose that effective immunosurveillance of senescent cells in salamanders supports their ability to undergo regeneration throughout their lifespan.

*For correspondence: maximina. yun@ucl.ac.uk

Competing interests: The authors declare that no competing interests exist.

## Introduction

Cellular senescence was previously identified as a process that permanently halts the proliferation of normal cells in culture following replicative exhaustion (*Hayflick and Moorhead, 1961*). It subsequently became clear that cellular senescence is a stress response that acts both in culture and in vivo to prevent proliferation of cells exposed to oncogenic stress, such as telomere attrition, and various types of DNA damage and oncogene insertions (*Serrano et al., 1997*; *Bodnar et al., 1998*; *d'Adda di Fagagna, 2008*). It therefore acts as an effective anti-tumourigenic mechanism (*Braig et al., 2005*; *Chen et al., 2005*; *Collado et al., 2005*). However, senescent cells can also have detrimental effects on biological processes. Long-term accumulation of senescent cells leads to disruption of tissue structure and function, possibly through the acquisition of a senescence-associated secretory phenotype (*Campisi, 2005*; *Campisi et al., 2011*). This is particularly relevant in the context of ageing, as in most species there is a marked accumulation of senescent cells with time (*Herbig et al., 2006*; *Wang et al., 2009*; *van Deursen, 2014*). Indeed, recent studies have uncovered a causative link between cellular senescence and age-related deterioration (*Baker et al., 2008*, *2011*, *2013*), underscoring the therapeutic benefits of targeting senescent cells (*Naylor et al., 2013*; *van Deursen, 2014*), and establishing cellular senescence as a hallmark of ageing (*Lopez-Otin et al., 2013*).

In mammals, the ability to regenerate tissues declines with age (*Sousa-Victor et al., 2014*). The decline in muscle regeneration with age has recently been linked to the loss of quiescent stem cells through senescence (*Sousa-Victor et al., 2014*). In contrast, other vertebrates, such as zebrafish and salamanders, are able to accomplish perfect regeneration of a wide range of complex structures throughout their lifespan (*Margotta et al., 2002*; *Azevedo et al., 2011*; *Eguchi et al., 2011*;

**eLife digest** As humans and other mammals get older, they become less able to recover from injury or repair damage to their tissues. This happens because mammalian cells gradually lose the ability to divide to produce new cells. This process is called senescence and it helps to prevent cancer by stopping old cells that are more likely to carry harmful mutations from replicating. However the link between senescence and many age-related declines in human health has led scientists to ask whether targeting senescent cells might be one way to treat age-related conditions.

Some organisms can regenerate their tissues throughout their lives; and creatures like salamanders are even able to re-grow limbs and organs if they are lost. Scientists are eager to learn how these animals are able to do this when humans are not, and answering this and related questions might help us to develop therapies that boost our ability to recover from injury or age-related diseases.

Yun et al. took a closer look at senescence in salamanders and unexpectedly found that a large number of senescent cells appeared in a salamander limb as it regenerates. But, by the time the limb had completely regrown, these senescent cells had disappeared. Further experiments revealed that when normal and senescent cells are implanted into a salamander the senescent cells also quickly disappear.

These findings suggest that senescent cells may possibly play a role in the regeneration process, and that salamanders have a system that can efficiently destroy these cells. Previous research had suggested that parts of the immune system, in particular cells called macrophages, help to eliminate senescent cells in some tissues. Yun et al. found that macrophages did accumulate around senescent cells in the regenerating limbs of living salamanders. And when a toxin was used to destroy the macrophages in some salamanders, the senescent cells were not cleared in the way they were in salamanders with active macrophages. Hence, macrophages are an essential part of the mechanism that eliminates senescent cells from salamander tissues.

This efficient mechanism for the elimination of senescent cells could explain how salamanders are able to maintain their ability to regenerate in spite of ageing. These findings also reveal the salamander as a model system that could be used to find new ways to target senescent cells, which could be eventually used in anti-ageing therapies.

*Itou et al., 2012*). It is therefore possible that these organisms exhibit mechanisms to curtail cellular senescence, thus allowing them to undergo indefinite rounds of regeneration. However, the occurrence and regulation of cellular senescence during regeneration have not been addressed in such regeneration-competent species. Here, we analysed the process of cellular senescence in salamanders. Our results demonstrate that cellular senescence is a recurrent process during salamander limb regeneration, and is subject to a highly efficient mechanism of macrophage-dependent elimination.

## Results

To investigate the occurrence of cellular senescence in the salamander system, we first adapted and validated reagents that allow the identification of senescent cells in cell culture and tissue sections. Cellular senescence is a mechanism that irreversibly arrests the proliferation of damaged or dysfunctional cells (*Shay and Roninson, 2004*; *Campisi, 2005*), and is triggered upon DNA insults such as UV irradiation (*Campisi, 2011*). We therefore exposed A1 salamander limb cells to UV, while stabilising the level of p53, which is required for the induction of senescence in certain mammalian contexts (*Figure 1A*). Up to 80% of the irradiated cells entered a state of cell cycle withdrawal within 12 days that, in contrast to quiescence, was accompanied by high levels of Gadd45 expression, permanent γH2AX foci, and high levels of senescence-associated β-galactosidase (SA-βgal), a hallmark of cellular senescence (*Dimri et al., 1995*; *Campisi, 2011*) (*Figure 1B–F*). The timing of SA-βgal induction in salamander cells is similar to that reported in mammalian fibroblasts undergoing stress-induced senescence (*Parrinello et al., 2003*).

Senescent cells are characterised by an irreversible cell cycle arrest. While proliferating or quiescent A1 cells re-enter the cell cycle upon serum stimulation (indicated by the incorporation of the

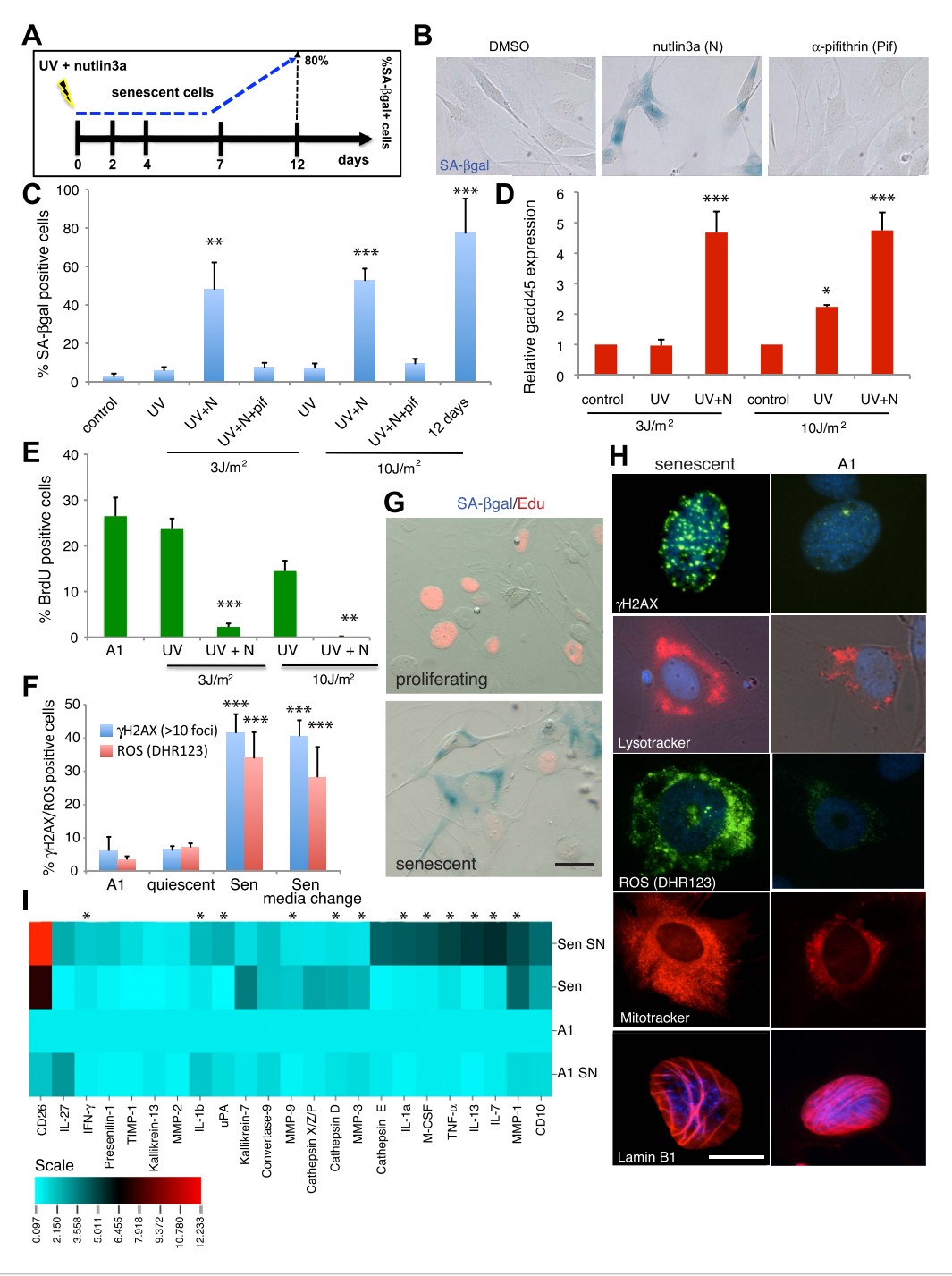

**Figure 1**. Characterisation of cellular senescence in salamander cells. (**A**) Protocol for the induction of cellular senescence in salamander cells. A1 cells are induced to undergo cellular senescence upon UV irradiation combined with p53 stabilisation (1 μM nutlin3a). SA-β-gal positive cells (blue) appear at 7 days post-induction (dpi) and represent 80% of the total population by 12 dpi. (**B**) SA-β-gal staining of UV (10J/m²) irradiated cells at 12 dpi in the presence of the indicated compounds. (**C**–**F**) Quantification of SA-β-gal (**C**), BrdU (**E**), DHR123 (ROS) and γH2AX foci (**F**) positive cells, or gadd45 expression relative to ef1-α (**D**) at 7dpi following the indicated UV doses/treatments (n = 4). (**G**) SA-β-gal/EdU co-staining following subculture of proliferating or senescent cells in growth medium supplemented with 5 μM EdU. No SA-β-gal⁺/EdU⁺ cells are found, indicating cell cycle withdrawal of SA-β-gal⁺ cells (n = 5). Scale bar: 50 μM. (**H**) The cells identified by the Sa-β-gal staining exhibit other hallmarks of senescence such as persistent γH2AX foci (γH2AX immunostaining), high levels of ROS (DHR123), and extended mitochondrial

*Figure 1. continued on next page*

*Figure 1. Continued*

(Mitotracker) and lysosomal (Lysotracker) networks. Lamin B1 is present at normal levels in senescent cells (representative images, n = 4). Scale bar: 50 μM. (**I**) Senescence-associated secretory phenotype analysis reveals similarities between salamander and mammalian senescent cells. Conditioned medium (SN) and whole cell extracts from A1 control (A1) or senescent (Sen) cells were analysed by cytokine/protease antibody arrays. The proteins whose levels vary significantly (p < 0.05) relative to A1 whole cell extracts are represented in the heat map. Asterisks indicate SASP factors upregulated in both salamander and mammalian senescent cells.

nucleotide analogue EdU), SA-βgal positive A1 cells do not, as demonstrated by double SA-βgal/Edu staining (*Figure 1G*). Furthermore, the cells identified by the SA-βgal staining exhibit other markers of senescence (*Campisi, 2005*) such as sustained production of reactive oxygen species (ROS), and extended mitochondrial and lysosomal networks (*Figure 1F,H*) relative to non-senescent parallel cultures. In addition, these cells acquire a secretory phenotype comprising a number of cytokines, signalling and matrix remodelling factors (*Figure 1I*). It is nothworthy that many of these factors, such as the interleukins IL-1a, IL-1b, IL-13, IL-7, IFNγ (Interferon gamma), TNF-α (Tumor Necrosis Factor alpha), M-CSF (Macrophage Colony-Stimulating Factor), the matrix metelloproteases MMP1, 3 and 9, uPA (urokinase-type Plasminogen Activator) and Cathepsin D, have been previously reported to integrate the senescence-associated phenotype (SASP) in various contexts of mammalian senescence (*Coppé et al., 2010*). Together, these data suggest that the regulation and properties of the senescent state are comparable in salamanders and mammals.

## Induction of cellular senescence during salamander limb regeneration

In order to determine whether cellular senescence occurs in vivo within normal and regenerating salamander tissues, we adapted the SA-βgal staining to detect senescent cells within salamander tissues and addressed whether senescence occurs during limb regeneration in an adult salamander, the newt (*Notophthalmus viridescens*). We found that there is a significant induction of cellular senescence during the intermediate stages of regeneration (mid-blastema and palette), but that the number of senescent cells diminishes subsequently (*Figure 2A,B*). Within the regenerating limb, senescent cells are found at the level of the amputation plane, including cartilage and muscle, and within the blastema, including fibroblast, monocyte-like cells, and epidermal glands (*Figure 2—figure supplement 1*). To corroborate that SA-βgal⁺ cells exhibit other hallmarks of cellular senescence in vivo, we performed SA-βgal/EdU stainings in regenerating or normal tissues following systemic EdU injections. We were unable to detect SA-βgal⁺/EdU⁺ cells under any condition examined (*Figure 2C*), suggesting that SA-βgal⁺ cells are withdrawn from the cell cycle, a characteristic of senescent cells.

Similar observations were made in another salamander species, the axolotl (*Ambystoma mexicanum*), both in mature animals as well as at early stages of development (*Figure 2D,E*). Interestingly, the induction and subsequent disappearance of senescent cells appear to be characteristic of regeneration, as no significant induction of senescence takes place during normal limb development (*Figure 2E*, *Figure 2—figure supplement 2*), in contrast to previous findings in amniotic limb development (*Banito and Lowe, 2013*; *Munoz-Espin et al., 2013*; *Storer et al., 2013*). These observations suggest that cellular senescence is a normal process during salamander limb regeneration, and is subject to dynamic regulation.

## Cellular senescence does not increase upon repetitive amputations or ageing

Salamanders have the remarkable ability to undergo multiple rounds of regeneration through their lives (*Eguchi et al., 2011*). Given the proportion of senescent cells induced during each regeneration cycle (7% of total cells at the mid-blastema stage), it seemed possible that senescent cells could accumulate following multiple rounds. However, we found that the percentage of senescent cells does not increase following sequential amputation–regeneration cycles. The tissues of adult newts that had been subjected to five regeneration cycles over 1.5 years did not show any accumulation of senescent cells compared to intact limbs (*Figure 2F*). This data suggests that elimination of senescent cells occurs during each round of limb regeneration.

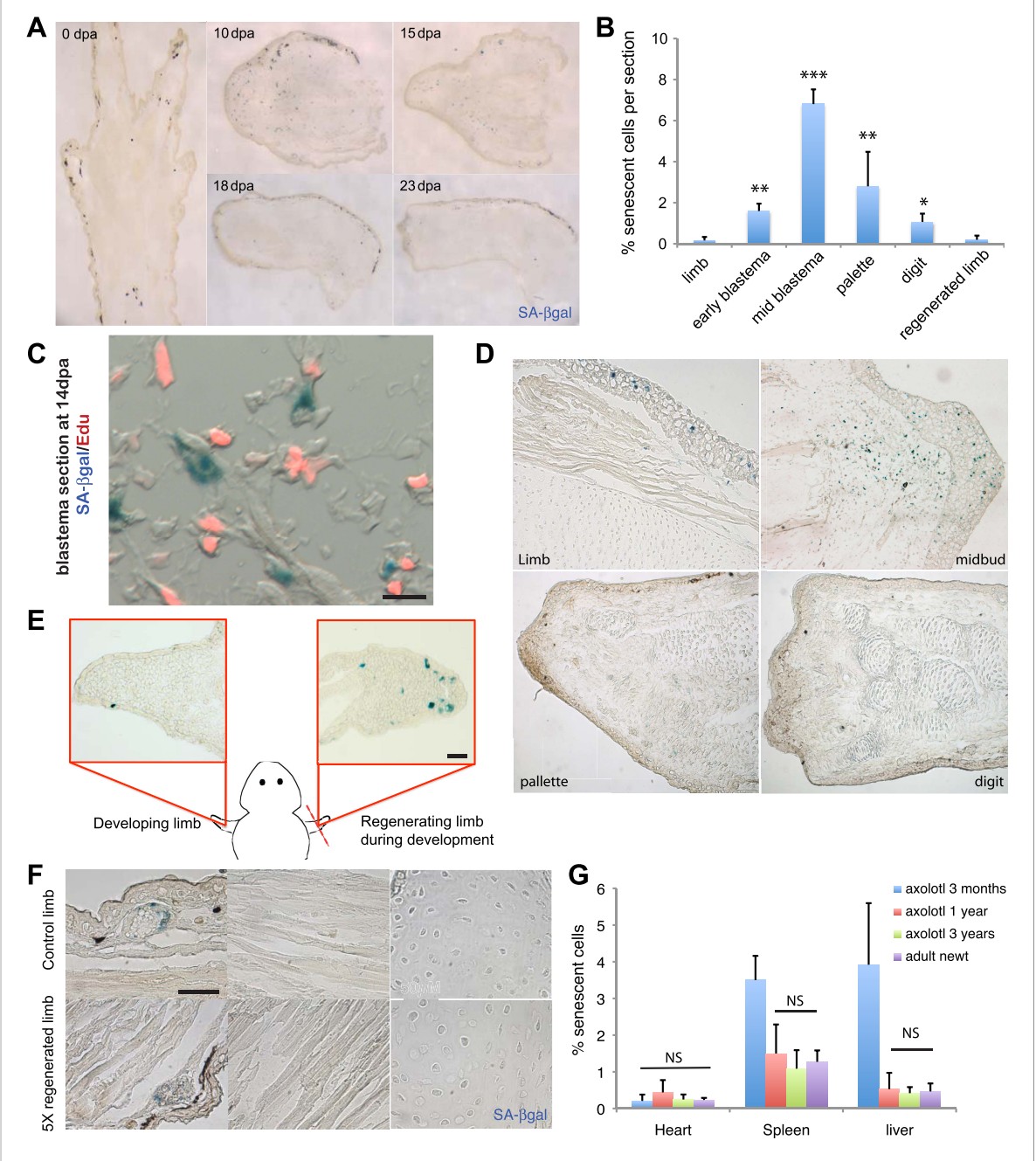

**Figure 2**. Induction of cellular senescence during salamander limb regeneration. (**A**) Induction and disappearance of senescent cells during salamander limb regeneration as indicated by SA-β-gal staining of newt tissues. dpa: days post amputation. (**B**) Quantification of senescent cells during key stages of limb regeneration following SA-β-gal/Hoescht staining (n = 8; *p < 0.05, **p < 0.01, ***p < 0.005). (**C**) Representative image of SA-β-gal/EdU co-staining of newt blastemas at 14 dpa, 48 hr after EdU administration. No SA-β-gal+/EdU+ cells are found (n = 6). Scale bar: 40 μM. (**D**) Induction and disappearance of senescent cells during axolotl limb regeneration as indicated by SA-β-gal staining (blue). dpa: days post amputation. Scale bar: 200 μm. (**E**) Cellular senescence is specifically induced during salamander limb regeneration, but not limb development. Representative SA-β-gal staining (n = 5) of a developing limb (left) and its contralateral counterpart following amputation and subsequent regeneration of the limb bud (right). Scale bar: 100 μm. (**F**) Repetitive amputation rounds do not lead to an increase in cellular senescence in regenerated limbs. Representative images SA-β-gal staining in a newt limb following 5 regeneration rounds compared to its contralateral intact limb (n = 6). Scale bar: 100 μm. (**G**) The percentage of senescent cells does not increase with ageing in salamander tissues. Quantification of senescent cells following SA-β-gal/Hoescht staining (n = 6). The percentage of senescent cells in tissues of mature axolotls (1 and 3 year-old) and adult newts is not significantly different (NS).

The following figure supplements are available for figure 2:

*Figure 2. continued on next page*

*Figure 2. Continued*

**Figure supplement 1**. Distribution of senescent cells during limb regeneration.

**Figure supplement 2**. Senescence is induced specifically during regeneration but not development of the salamander limb.

**Figure supplement 3**. Cellular senescence does not increase with ageing in salamanders.

---

The proportion of senescent cells within tissues is known to increase as the organism matures in most species studied so far, including mammals (*Herbig et al., 2006*; *van Deursen, 2014*). Remarkably, we found that the percentage of senescent cells in heart, spleen and liver does not increase in older salamanders (*Figure 2G*, *Figure 2—figure supplement 3*); the latter two organs are known to accumulate senescent cells in mammals as they age (*Herbig et al., 2006*; *van Deursen, 2014*). Together, these findings raise the possibility that active mechanisms of senescent cell clearance operate within normal and regenerating salamander tissues.

## Efficient surveillance of senescent cells in adult salamander tissues

To test this possibility, we implanted one thousand senescent or normal cells (labelled with nuclear GFP expressed from an integrated retrovirus) within newt limb tissues and analysed their persistence over time (*Figure 3A*). After an initial non-specific loss of cells following implantation, the normal cells (A1 GFP+) persist for at least 40 days and contribute to structures within the limb tissues (*Figure 3A,B*). In contrast, 80% of the implanted senescent cell population is cleared from the limb tissues within 2 weeks post-implantation (*Figure 3A,B*), consistent with a half-life of 9 days (calculated from the exponential equation corresponding to the senescent dataset, after subtraction of the initial loss observed in normal cells), comparable to the rate of clearance of senescent cells during regeneration (*Figure 2*). Although this experiment suggests that senescent cells are being cleared from salamander tissues, it cannot exclude the possibility that some cells may reverse their senescence state. To distinguish between these two possibilities, we performed the same experiment using retrovirally labelled nGFP senescent cells (*Figure 3B* and *Figure 3—figure supplement 1*). The rate of clearance of nGFP+ senescent cells corresponds to the rate of clearance of unlabelled senescent cells, supporting the clearance hypothesis. In contrast, the implantation of a population of nGFP cells exposed to a UV dose of 1J/m2, 10 times lower than the dose required for senescence induction (*Figure 3—figure supplement 1*), results in persistence of the implanted cells (*Figure 3B*). These data suggest that salamander tissues have a significant capacity for senescent cell clearance.

Interestingly, the implantation of a 1:1 mixture of senescent and normal cells does not result in the persistence of the normal cells, but in the clearance of both populations (*Figure 3B*, *Figure 4A,B*), albeit with a delay when compared to the clearance of purely senescent cells. It has been recently shown that senescent cells are able to induce senescence in their neighbours, a phenomenon termed 'senescence bystander effect' (*Nelson et al., 2012*; *Acosta et al., 2013*). We found that this property is also present in senescent salamander cells. Cellular senescence can be induced in normal cells by co-culture with senescent cells (*Figure 4C*), or following incubation with purified media from senescent populations (*Figure 4D*), supporting the idea that senescence can be transmitted in a paracrine fashion. In addition, implantation of the 1:1 mixture of senescent + normal cells results in the induction of senescence in normal cells (*Figure 4E,F*), providing an explanation for their eventual clearance. Together, these results suggest that active mechanisms identify and target senescent cells for clearance within salamander tissues.

## Macrophages are a critical part of the senescent cell clearance mechanism

A series of elegant experiments have recently described a role for the immune system in the surveillance and clearance of p53 or oncogene-induced senescent hepatocytes in mice (*Xue et al., 2007*; *Kang et al., 2011*; *Lujambio et al., 2013*). In addition, recent evidence suggests that there is an extensive recruitment of immune cells, and in particular macrophages, to the regenerating limb in salamanders (*Godwin et al., 2013*). Hence, we decided to investigate if the macrophage was part

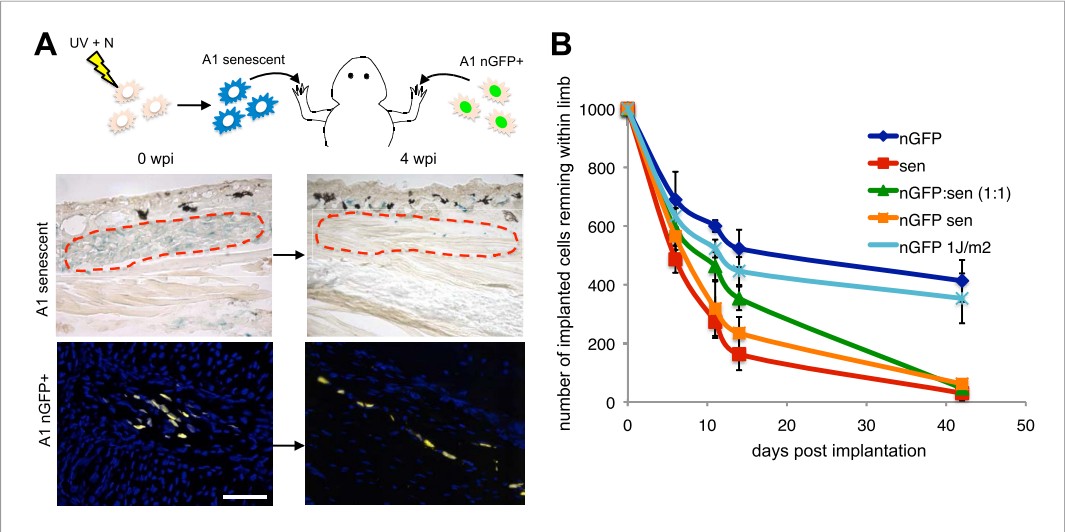

**Figure 3**. Active mechanisms of senescent cell clearance operate in salamander tissues. (**A**) Schematic representation of the implantation experiment. 1000 UV-induced senescent or nGFP+ non-senescent cells were implanted within the left or right newt forelimbs, respectively, and analysed at different weeks post-implantation (wpi) by SA-b-gal staining or immunofluorescence (below). Note the complete clearance of senescent cells at 4wpi. Scale bar: 100 μm. (**B**) Dynamics of senescent cell clearance within adult newt limbs as described in **A** as shown by a quantification of total cells remaining within entire limbs at different days post implantation (dpi, n = 12). In addition to senescent and nGFP+ control cells, the dynamic of cell clearance was evaluated following implantation of nGFP+ senescent cells, nGFP+ UV-irradiated (1J/m2) cells and a 1:1 mixture of nGFP+ control and senescent cells (nGFP:sen).

The following figure supplement is available for figure 3:

**Figure supplement 1**. Distribution of senescent cells during limb regeneration.

of the senescence clearance mechanism. In vivo labelling using TMR-dextran, which is specifically internalised by macrophages (*Figure 5A–C*), as well as monocyte specific α-naphthyl acetate esterase staining (*Figure 5D*), reveal that macrophages and senescent cells are found in close proximity within regenerating limbs, suggesting cell-to-cell contact and engulfment in some cases. To determine whether macrophages mediate the clearance of senescent cells, we took advantage of an efficient method for macrophage depletion, based on the intravenous administration of selectively toxic, clodronate salt-filled liposomes (clodrosomes). These are internalised specifically by macrophages via phagocytosis (*van Rooijen and Hendrikx, 2010*) and have been shown to operate effectively in salamanders (*Godwin et al., 2013*). We found that treatment with clodrosomes is able to reduce the macrophage population by fivefold, compared with control DiI liposomes (*Figure 6A*). As with the TMR-dextran, these liposomes are specifically internalised by macrophages, as shown by their incorporation into cells expressing F4/80, a specific macrophage marker (*Morris et al., 1991*) (*Figure 6B*). We therefore used this system to induce systemic macrophage depletion in salamanders prior to the implantation of senescent or normal nGFP+ cells in contralateral regenerating limbs, and followed the fate of the implanted cells with time (*Figure 6C*). At 20 hr post implantation, we observed a strong accumulation of DiI-labelled macrophages in the vicinity of senescent cells (*Figure 6D–F*). In contrast, macrophage recruitment to areas of normal cells (control) was not significantly different from their recruitment to other areas of the limb mesenchyme (*Figure 6D,F*). At 2 weeks post implantation, the normal cells remained within the regenerating limbs (*Figure 6G*). However, the senescent cells were cleared in animals injected with control DiI liposomes, but remained in animals whose macrophages had been depleted (*Figure 6G*). This demonstrates that the macrophage is critical for clearance of senescent cells, and constitutes the first evidence for senescence surveillance mechanisms that operate during normal regeneration.

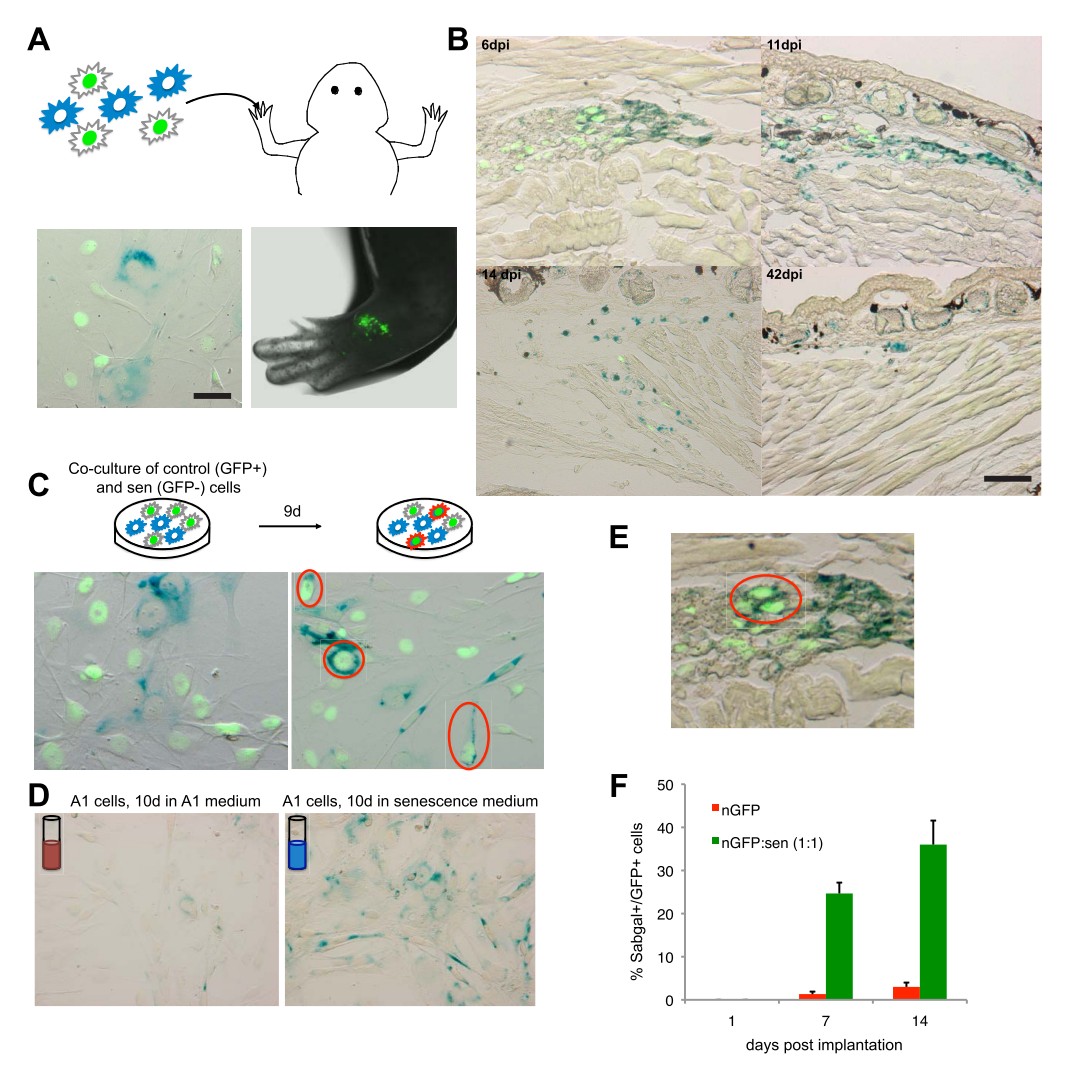

**Figure 4**. A senescence bystander effect in salamander cells and tissues. (**A**) Schematic representation: implantation of a 1:1 mixture of senescent and non-senescent (nGFP+) salamander cells within limb tissues. Scale bar: 50 μM. (**B**) Both senescent and non-senescent cell populations are cleared from newt limb tissues following implantation. Scale bar: 100 μM. (**C**) Co-culturing of senescent and non-senescent (GFP+) cells promotes cellular senescence in the non-senescent population -circled areas-. Scale bar: 50 μM. (**D**) Incubation of proliferating salamander cells with growth media derived from senescent (blue) but not normal (red) cells induces cellular senescence. Scale bar: 100 μM. (**E**) Co-implantation of normal and senescent cells leads to the induction of senescence in the non-senescent (GFP+) cell population -circled areas-. (**F**) Percentage of SAβgal+ nGFP+ cells when co-implanted with (nGFP-) senescent or normal cells at different times post implantation. (n = 3, **p < 0.01).

## Discussion

It is clear that salamanders possess a rapid and efficient mechanism to recognise and clear senescent cells that either arise endogenously, or are introduced from culture. Our study demonstrates that a robust macrophage-dependent surveillance mechanism operates in normal and regenerating tissues of adult salamanders, and this allows them to circumvent the negative effects associated with the long-term accumulation of senescent cells, such as the disruption of tissue structure and function (*Campisi, 2005*; *Rodier and Campisi, 2011*). These surveillance mechanisms are particularly significant for limb regeneration because there is a notable induction of cellular senescence during this process. Consistent with this, recent reports show that systemic macrophage depletion during

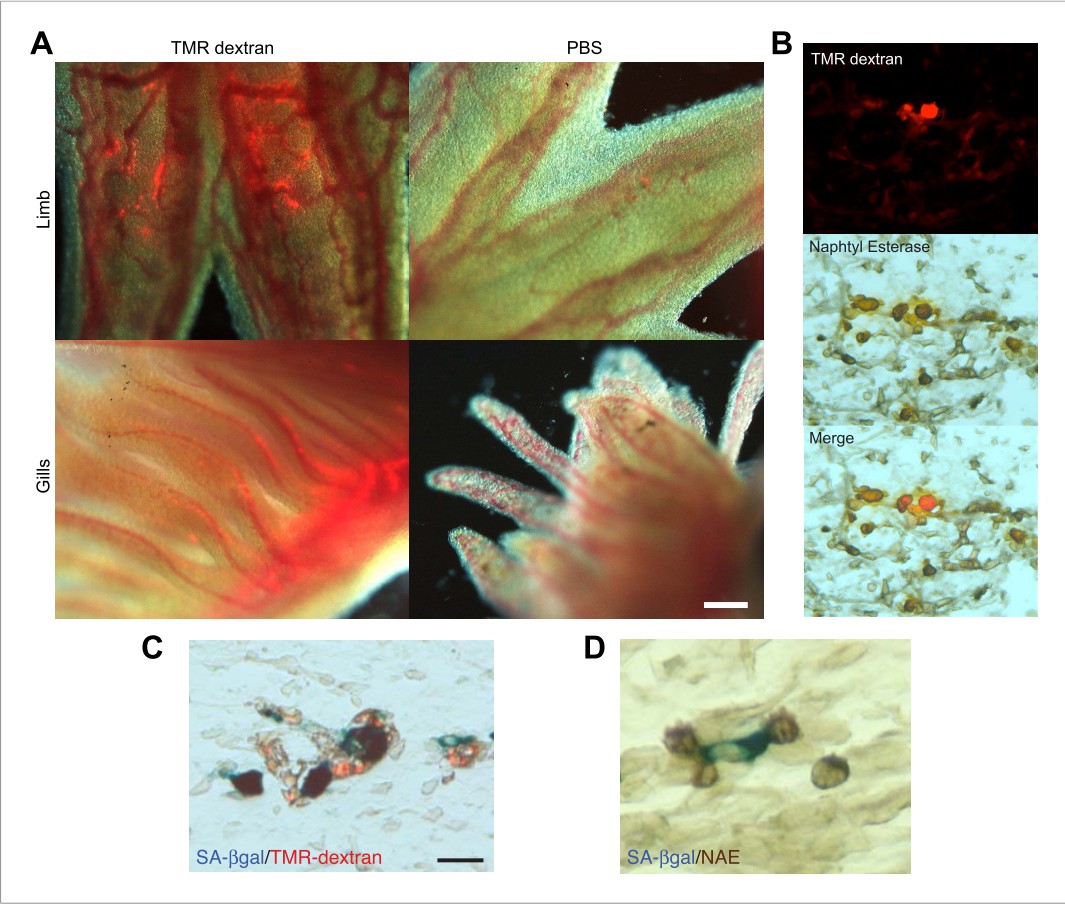

**Figure 5**. Macrophages are recruited to senescent cells within salamander tissues. (**A**) Representative images of axolotl limbs and gills, 24 hr after i.v. injection of TMR-dextran -left panels- or PBS -right panels-. Scale bar: 200 μM. (**B**) TMR-dextran incorporating cells stain positive for the macrophage marker α-napthtyl esterase (NAE). (**C**, **D**) Macrophages (TMR+ -red- in **A**; NAE+, brown in **B**) are recruited to sites of senescent cells (Sa-β-gal+, blue) within regenerating limb tissues. Scale bar: μm.

salamander limb regeneration leads to defects in this process (*Godwin et al., 2013*). This has also been observed during zebrafish fin regeneration (*Petrie et al., 2014*). Furthermore, cellular senescence has been recently shown to contribute to the decline in muscle regenerative ability upon ageing in mammals (*Wang et al., 2009*; *Baker et al., 2011*; *Sousa-Victor et al., 2014*). In light of these observations, we propose that the effective immune clearance of senescent cells in salamanders supports their ability to undergo multiple rounds of regeneration through their lifespan (*Figure 7*).

The mechanisms underlying the induction of senescent cells and their clearance, for example the involvement of other immune cell types in their surveillance, constitute an important outstanding issue. The analysis of these problems has been limited by lack of reagents, as well as cellular and molecular tools for in vitro modelling of senescent-immune cell interactions as well as gain or loss-of-function approaches to evaluate induction and clearance of endogenous senescent cells. Further progress in this area will enable the exploration of these important matters.

Nevertheless, it is interesting that we find a consistent induction of cellular senescence during regeneration, suggesting that this could be integral to the mechanism. Senescent cells could play a positive role during regeneration, a proposal that is consistent with recent findings that senescence contributes to tissue remodelling during mouse development (*Munoz-Espin et al., 2013*; *Storer et al., 2013*; *Campisi, 2014*) and wound healing (*Krizhanovsky et al., 2008*; *Jun and Lau, 2010*; *Demaria et al., 2014*). Alternatively, senescent cells could contribute to regeneration indirectly, by promoting the efficient recruitment of macrophages, since the latter have been shown to promote

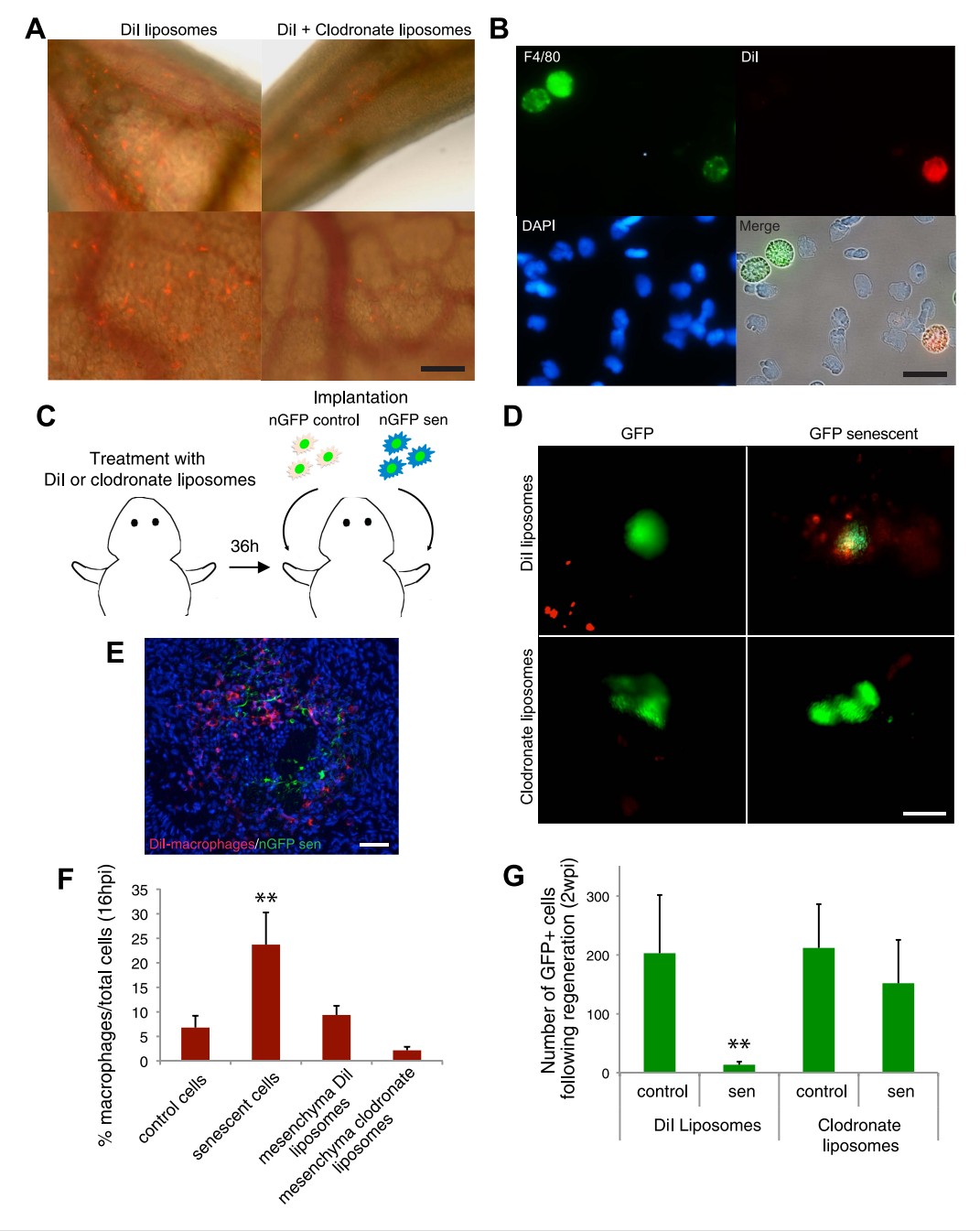

Figure 6. Macrophages mediate the efficient clearance of senescent cells during salamander limb regeneration. (A) A clodronate-dependent system for effective macrophage depletion in salamanders. Representative images of axolotl limb tissues 36 hr post i.v. injection with fluorescent DiI-liposomes (red) or DiI + clodronate-liposomes. Scale bar: 1000 μm. (B) DiI-liposomes are incorporated specifically by macrophages. Blood cells were extracted from treated animals and stained with antibodies against the specific macrophage marker F4/80. Scale bar: 50 μM. (C) Schematic representation of the implantation experiment. Axolotls with limbs at the midbud stage of regeneration were injected i.v. with either DiI-liposomes or DiI + clodronate(clodro)- liposomes, 36 hr prior to implantation of normal or senescent nGFP+ cells into contralateral regenerating limbs. Liposome treatments were maintained for 2 weeks. (D) Macrophage (DiI, red) recruitment to areas of senescent cells 12 hr following implantation of normal and senescent GFP+ cells within regenerating axolotl limbs. Scale bar: 1000 μm. (E) DiI-labelled macrophages (red) are recruited to sites of implanted nGFP+ senescent cells (green) within regenerating limbs at 16 hs post implantation. Scale bar: 100 μm. Note that in the DiI-liposome treated animals

*Figure 6. continued on next page*

*Figure 6. Continued*

macrophages are recruited to sited of senescent but not normal cells. (**F**) Quantification of macrophage (DiI-labelled) recruitment to implantation sites at 16 hpi. Note macrophage depletion induced by clodronate-liposome treatment. (**G**) Clearance of senescent cells is impaired upon macrophage depletion. Quantification of GFP+ cells after 2wpi in DiI-liposome or clodronate-liposome treated animals.

tissue remodelling and patterning in various contexts, including regeneration (*Lobov et al., 2005*; *Qian and Pollard, 2012*; *Petrie et al., 2014*). In either case, the senescence-associated secretory phenotype is likely to play an important role. These are important issues that merit further examination.

It has recently been suggested that targeting senescent cells could lead to therapeutic strategies for age-related pathologies (*Naylor et al., 2013*; *van Deursen, 2014*). Here, we identify an animal with an efficient mechanism for surveillance of senescent cells operating through adulthood. Analysis of this mechanism could lead to the identification of novel therapeutic targets for the amelioration of age-related disorders and extension of healthspan.

## Methods summary

Adult newts (*N. viridescens*) and axolotls (*A. mexicanum*) were used in this study, in compliance with the Animals (Scientific Procedures) Act 1986, approved by the United Kingdom Home Office. SA-β-gal activity was determined in cultured cells or tissues fixed with 0.5% glutaraldehyde, using the SA-βgal kit (Cell Signalling) according to the manufacturer's instructions. Immunofluorescence staining, EdU and BrdU labelling of tissues or cultured cells were performed according to standard protocols. Fluorescence and bright-field imaging was performed using a Zeiss Axioscope (Zeiss).

## Materials and methods

### Animal husbandry and procedures

Procedures for care and manipulation of all animals used in this study were performed in compliance with the Animals (Scientific Procedures) Act 1986, approved by the United Kingdom Home Office.

Adult newts (*N. viridescens*) were obtained from Charles Sullivan and Co. (Tennessee, USA) and maintained as described elsewhere (*Ferretti and Brockes, 1988*). Axolotls (*A. mexicanum*) were obtained from Neil Hardy Aquatica (Croydon, UK) and maintained in individual aquaria at approximately 18°C. Newts and axolotls were anesthetised in 0.1% tricaine prior to amputation at the mid-humerus level. Animals were allowed to regenerate at 20°C. For repetitive amputation–regeneration cycles,

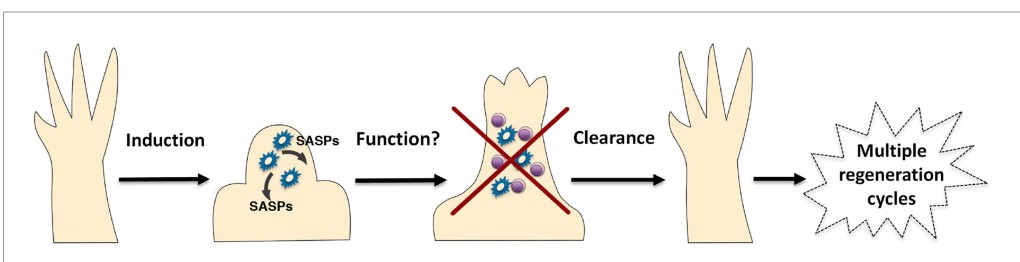

**Figure 7**. A simplified model for the induction and clearance of senescent cells during salamander regeneration. Salamander cells are induced (by as yet unidentified stimuli) during regeneration and accumulate within the blastema. They produce a range of secreted molecules (SASPs) which affect their microenvironment and could contribute to different aspects of the regeneration process such as matrix remodelling, vascularisation, growth and patterning. As regeneration proceeds to more advanced stages, senescent cells are cleared by an efficient mechanism of macrophage dependent immunesurveillance, which results in a regenerated limb devoid of senescent cells. This allows for the possibility of multiple rounds of regeneration through lifespan as well as avoiding the disadvantages of senescent cell accummulation.

animals were allowed to achieve full limb regeneration prior to re-amputation. Axolotl and newt regeneration stages were defined as previously described (*Iten and Bryant, 1973*; *Tank et al., 1976*).

## Cell culture

A1 cells were previously derived from newt limb mesenchyme (*Ferretti and Brockes, 1988*). A1 nGFP+ cells were obtained by stable transfection with a pseudotyped retroviral VSV-G vector as described below. Cells were grown on 0.75% gelatin-coated plastic dishes in MEM (Gibco, UK) supplemented with 10% heat-inactivated foetal calf serum (FCS, Gibco), 25% $H_2O$, 2 nM L-Glutamine (Gibco), 10 µg/ml insulin (Sigma, St Louis, MO) and 100U/ml penicillin/streptomycin (Gibco) in a humidified atmosphere of 2.5% $CO_2$ at 25˚C. Cell subculture was performed as previously (*Lo et al., 1993*). Human 293GP cells (*Burns et al., 1993*) were propagated in DMEM (Gibco) supplemented with 10% heat-inactivated foetal calf serum (FCS, Gibco).

## Generation of nGFP+ A1 cells using a pseudotyped retroviral vector

Pseudotyped retroviral vectors were generated essentially as described (*Yee et al., 1994*). 293GP cells containing the gag and pol MoMLV genes were cotransfected by standard calcium precipitation with the retroviral vector LZRSpBMN-Z—nGFP (encoding an eGFP protein with a nuclear localisation signal between the BglII and BamHI sites) and pCMV-VSV-G (an expression plasmid for Vesicular Stomatitis Virus protein G) in order to generate pseudotyped retrovirus. Medium was collected at 48, 60 and 72 hr post transfection, filtered through a 0.45 mm filter and concentrated 100 fold by ultracentrifugation at 25,000 rpm, 1.5 hr. Stock titers were determined by infecting QT6 cells. A1 cells were grown to 30% confluence and infected with pseudotyped retrovirus stocks ($1–2 \times 10^6$ PFU) in the presence of 8 µg/ml polybrene (Sigma). After overnight infection, cells were incubated in fresh culture medium with 2 mg/ml G418 until reaching confluence.

## Senescence induction in cell culture

In order to induce cellular senescence in cultures, cells were washed in amphibian PBS (A-PBS, PBS plus 25% dH2O) and exposed to 1, 3 or 10J/m² of UV irradiation (UV Stratalinker) in 1 ml A-PBS and incubated in growth medium, supplemented with 1 µM (−)nutlin3a (Cayman, Ann Arbor, MI) in a humidified atmosphere of 2.5% $CO_2$ at 25˚C. Nutlin3a was added to the growth medium in 1 µM increments every 24 hr for the duration of the experiment. Growth media was changed after 1 week. Cells were then subcultured and senescence induction evaluated using various methods described throughout this manuscript.

## SA-βgal staining

SA-β-gal activity was determined in 0.5% glutaraldehyde fixed cell culture cells using the SA-βgal kit (Cell Signalling, Danvers, MA) according to manufacturer's instructions. For detection of SA-β-gal in animal tissues, salamander embryos or adult tissues were fixed for 30 min or 1 hr respectively in 0.5% glutaraldehyde, washed 3 times in PBS. For whole mount staining, salamander embryos or adult tissues (heart, spleen, liver) were stained O/N using the SA-β-gal kit (Cell Signalling) according to manufacturer's instructions, washed in PBS, fixed in 4%PFA for 4 hr and embedded in Tissue Tek-II. Whole-mount samples were then cryosectioned, washed in PBS and mounted in glycerol for imaging.

Adult intact or regenerating limbs were cryosectioned in Tissue Tek-II, washed in PBS for 15 min and stained O/N using the SA-β-gal kit (Cell Signalling) according to manufacturer's instructions.

## Senescence-associated secretory phenotype analysis

Culture medium was cleared by centrifugation and concentrated using Ultracel-3K Amicon Ultra centrifugal tubes (Merck Millipore Ltd, Germany). The corresponding cell extracts were prepared by washing cells in PBS and resuspending them in ice-cold 1% NP40, 150 mM NaCl, 50 mM Tris–HCl, pH 7.5, 1 mM NaVO₄ and Protease Inhibitor Cocktail (Sigma), incubating for 30 min at 4˚C and clearing the debris by centrifugation. Concentrated media and their corresponding cell extracts were analysed using the Proteome Profiler Human Protease and Cytokine Array Kits (R&D systems, Minneapolis, MN) as per manufacturer's instructions. Antibody arrays were developed in an Odyssey scanner (LI-COR, Lincoln, NE), and protein levels quantified using Image Studio software. One Matrix CIM heat maps were built using CIMminer (NCI-NIH, USA).

## Cryosectioning

For the analysis of DiI liposome incorporation and NAE or NacdE staining of salamander limbs following cell implantation, regenerating or intact limbs were collected by amputation, fixed in 4% ice-cold paraformaldehyde (PFA) for 16–18 hr at 4°C, washed twice in PBS and embedded in Tissue Tek-II. The blocks were serially sectioned longitudinally in a cryostat (Leica, UK) at 12 µm. Sections were collected in Superfrost slides and stored at −30°C until use.

## EdU incorporation assays

To determine the percentage of cells in S-phase, proliferating, quiescent or senescent A1 cells were sub-cultured and incubated in growth medium supplemented with 5 µM 5-ethynyl-2′-deoxyuridine (EdU) for 24 hr. Cells were then fixed in glutaraldehyde and SA-β-gal staining performed as described. Next, cells were fixed in 4%PFA for 10 min and stained using Click-iT Edu Alexa Fluor 594 Imaging kit (Life Technologies, UK) according to manufacturer's instructions.

To detect EdU incorporation in salamander tissues, 10 mM EdU (20 µl per animal) were administered to regenerating newts by i.p. injection. After 48 hr, blastemas were collected, cryosectioned and stained to determine SA-β-gal activity as described. Sections were fixed in 4% PFA for 10 min and EdU incorporation determined using Click-iT Edu Alexa Fluor 594 Imaging kit (Life Technologies).

## Cell implantations

Cellular senescence was induced in cultured A1 cells by UV irradiation as described above. Following verification by O/N SA-β-gal staining, senescent cells were trypsinised, spun at 850 rpm for 10 min and the resulting pellet resuspended in A-PBS. Cells were then transferred to a 10 µl Hamilton syringe (Hamilton, Reno, NV) with a $30^{1/2}$ g, 45° tip needle (Hamilton) attached to a micromanipulator. 1000 cells were injected within the mesenchymal tissues of intact or regenerating (midbud stage) limbs of newts or axolotls. The animals were allowed to recover on a wet tissue for 10 min before transfer to water. Successful implantation was verified by fluorescence stereomicroscopy using a Zeiss Axioscope (Zeiss, Germany).

## In vivo macrophage labelling

17 cm axolotls (3 year old animals at a reproductive stage) were anaesthetised in 0.1% Tricaine for 20 min, prior to i.v. microinjection of 20 µl DiI or clodronate liposome solutions (Encapsula Nano Sciences, Brentwood, TN) as previously described (*Godwin et al., 2013*). For long-term depletion experiments, animals were injected every other day 3 times, with a resting period of 3 days before the next injection rounds. The same procedure was applied for macrophage labelling using 20 µl 2MDa tetra-methylrhodamine (TMR) labelled dextran (Invitrogen, UK). All injections were performed 36–48 hr before cell implantation or analysis. Successful compound uptake by macrophages was verified by fluorescence stereomicroscopy using a Zeiss Axioscope (Zeiss).

## Cytochemistry

Granulocytes -including neutrophils- were detected by specific staining of tissue cryosections with the naphtol AS-D chloroacetate esterase kit (Sigma–Aldrich), while monocytes/macrophages were identified using the α-naphtyl acetate esterase (NAE) kit (Sigma–Aldrich), as previously described[18].

## Cell immunofluorescence

For staining of cultured cells, these were fixed in 2% PFA for 1 min, followed by a 5-min incubation in cold 100% methanol and processed as described elsewhere (*Yun et al., 2014*). Fixed cell samples were incubated overnight with the following primary antibodies: anti-BrdU (Sigma; 1:3000), anti-γH2AX (Upstate, UK; 1:1000), anti-laminB1 (Abcam, UK; 1:500), and anti-F4/80 (ABD Serotec, UK; 1:500).

In all cases, anti-mouse or anti-rabbit AlexaFluor488 and AlexFluor594 antibodies (Invitrogen; 1:1000) were used for secondary staining. Hoechst 33258 (2 µg/ml) was used for nuclei counterstaining.

For detection of ROS production, cells were incubated with 10 µM DHR123 (Life Technologies, Invitrogen) for 2 hr at 25°C, fixed and mounted in DAKO fluorescent mounting media.

Mitochondrial and Lysosomal networks were visualised following incubation of cells with 200 nM MitoTracker red FM and 500 nM LysoTracker red DD99 (Life Technologies, Invitrogen) respectively for 1 hr at 25°C.

Samples were observed under a Zeiss Axiskop2 microscope and images were acquired with a Hamamatsu Orca camera using Openlab (Improvision) software. Whenever comparative analyses between samples were performed, all images were acquired with identical camera settings and illumination control. Image processing (contrast enhancement) was equally applied to all matched experimental and control samples using Openlab software.

### BrdU analysis

Cells were labelled for 2 hr by adding 1 µl/ml 5-bromo-2′-deoxyuridine (BrdU) to the growth media. Following the corresponding incubation period, cells were fixed in 4% paraformaldehyde for 1 min at RT followed by 100% methanol for 5 min, and stained for bromodeoxyuridine as previously described (*Yun et al., 2013*).

### Quantitative RT-PCR

RNA was isolated from salamander tissue culture cells using Tri Reagent (Sigma) and random primed cDNA synthesised using Superscript II (Invitrogen). Gene expression was determined by quantitative real time PCR using the following primers: Newt Gadd45-β fwd (AGGGCACAGGAAAGAAGATG); Newt Gadd45-β rev (TCATTGTCGCAGCAGAAGG); Newt L-27 fwd (TACAACCACTTGATGCCA); Newt L-27 rev (CAGTCTTGTATCGTTCCTCA). Rt-PCR was carried out using iQ SYBR Green supermix (Bio-rad, Hercules, CA) on a Chromo 4 instrument running Opticon 3 software (Bio-rad). All reactions were run in triplicate and at least 2 independent RNA preparations were analysed for each sample.

### Statistical analysis

Newts and axolotls in each sample group were randomly selected. Sample group size (n) is indicated in each figure legend, while all experiments were carried out in at least three biological replicates. Statistical analyses were performed with Prism 4.0 software and unpaired two-tailed t tests were applied unless otherwise stated. Paired two-tailed t tests were carried out to analyse RT-PCR experiments.

## Acknowledgements

We thank Anna Czarkwiani for technical advice, Anoop Kumar for generating the A1 nGFP cell line, and Roger Franklin, Phillip Gates and Andrew Osborne for helpful comments on the manuscript. This work was supported by a Wellcome VIP award to MHY, an MRC studentship to HD and an MRC programme grant to JPB.

## Additional information

### Funding

| Funder | Grant reference | Author |
| --- | --- | --- |
| Medical Research Council | MRC Programme Grant | Jeremy P Brockes |
| UCL - Wellcome Trust | VIP award | Maximina H Yun |
| Medical Research Council | Studentship | Hongorzul Davaapil |

The funders had no role in study design, data collection and interpretation, or the decision to submit the work for publication.

### Author contributions

MHY, Conceived the project, Designed and performed experiments, Analysed/interpreted data, Wrote and revised the manuscript; HD, Performed experiments and analysed/interpreted data; JPB, Designed experiments, Analysed/interpreted data and revised the manuscript

### Ethics

Animal experimentation: All procedures for care and manipulation of adult newts (*Notophthalmus viridescens*) and axolotls (*Ambystoma mexicanum*) were performed in compliance with the Animals (Scientific Procedures) Act 1986, approved by the United Kingdom Home Office and University College London (Institutional License: 70-2716). All surgical procedures were carried out under anesthesia (0.1% Tricaine) followed by treatment with analgesics (Butorphanol tartrate). Every effort was made to minimise suffering.

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
