## [Decision Letter]

Thank you for sending your work entitled “Recurrent turnover of senescent cells during regeneration of a complex structure” for consideration at *eLife*. Your article has been favorably evaluated by Fiona Watt (Senior editor), three reviewers, and a member of our Board of Reviewing Editors. The reviewers raised a number of concerns that will have to be addressed before publication can be envisaged. This will require further experimentation as well as modifications to the text.

The Reviewing editor and the reviewers discussed their comments before we reached this decision, and the Reviewing editor has assembled the following comments to help you prepare a revised submission.

The finding that senescent cells are present in regenerating salamander tissue and that they are cleared by macrophages, with lack of persistence both after repeated injury or on ageing, is of interest, although cell senescence and macrophage clearance have been reported previously in the context of wound healing. To consolidate the results and provide more mechanistic insight the authors should address the following issues:

1) The authors use the concept that since there are no EdU positive cells that are also SA-βgal positive, then the senescent cells must be cleared by macrophages. However, the investigators fail to mention or consider that if there was a SA-βgal positive cell that incorporated EdU then the SA-βgal staining would go away. This needs to be addressed very carefully, since this is a central point in the manuscript.

Related to this, it is not clear from the text if the transplanted senescent cells are also expressing a fluorescent marker; it seems not. If not, it would be better to have experiments tracking the senescent cells with an independent marker such as RFP. As it stands, although unlikely, cells could be reversing the senescence and losing B-gal expression rather than being cleared from the blastema.

2) The authors should consider depleting macrophages from salamanders using the technique they described, then induce regeneration and provide quantitative data on βgal positive cells with time in a real animal. Do the numbers of senescent cells still disappear in macrophage depleted animals similarly to salamanders with macrophages?

What happens in vitro when salamander A1 senescent cells are mixed with macrophages?

3) Many immune processes are dependent on macrophages and there are several different cells types that could have been examined. The lack of molecular mechanisms of the pathways from senescent cells to them being cleared is a serious weakness.

That both normal and senescent cells are cleared but with a different time course suggests the mechanisms of clearance may be more complex than the investigators claim.

4) It is disappointing that the investigators did not attempt to identify what is secreted by the senescent cells? Is it the same SASP factors as demonstrated in mammals? This would have added some mechanistic information to the study.

Also, nothing is detailed about the mechanism of how senescent cells can induce senescence in normal cells.

5) The role of paracrine-senescence is poorly investigated and does not contribute significantly to the paper. The fact that it can be induced is clear, as has been reported in a number of other papers (Acosta et al., Nature Cell Biol, 2013 reference is missing). And this could help to explain how the co-injection of senescent cells with normal cells favors the removal of the normal. However, it could as easily be that the senescent cells simply recruit immune cells to the site of injection, which then remove any exogenous cells. Figure 4 suggests that paracrine senescence may contribute, but this should be quantified to show this is the mechanism.

Furthermore, it is claimed that senescent cells are found in clusters in the spleen because of a paracrine-senescence effect. It is more likely that senescent cells injected in one site are being removed to the spleen, and this is why they appear in “clusters”. Such claims should be experimentally validated.

6) The use of UV as an irreversible senescence inducer is not robust. UV treatment of mammalian cells (depending on the dose) results in cell repair or death. UV induces p53 and a variety of different types of DNA strand breaks, and depending on the dose, many cells can repair and survive. Almost nothing is mentioned about a dose curve for these experiments. If the investigators use a dose that induces senescence but is reversible (low doses) do the cells clear similarly to normal or senescent cells.

7) It appears that senescence can only be induced in salamander cells by the stabilization of p53, but this is completely omitted from the Discussion. This is confusing, given that the same group has recently shown that down regulation of p53 correlates with the timing of senescence induction (Yun et al., PNAS 2013). In addition, nutlin treatment impairs limb regeneration if treated at the blastema stage, but increased it at the regeneration phase.

Given the importance of p53 in the regulation of senescence and their previous paper, this needs more detailed explanation.

8) The fact that there is an increase in senescence with wounding is well established in other studies (Jun and Lau, Nature Cell Biology, July 2010; Krizhanovsky et al., Cell, 2008), as is the fact that senescent cells are cleared by macrophage-mediated removal (Xue et al., Nature, 2007; Krizhanovsky et al., Cell, 2008; Kang et al., Nature 2011). These papers should be cited.

9) Most figures need better explanation in the text, describing exactly what was done, and in enough detail to follow. The figure legends similarly need more detail.

Many figures and experiments also lack quantification and controls. While the result can be taken at face-value as a description of what was observed, more detail would make the data much more convincing.

E.g. Figure 1 needs control staining of non-senescent cells to show these markers of senescence are increased under these conditions.

Figure 2, it is quite difficult to see the senescent cells and where they are located. Higher resolution images, with some reference to localization/distribution are needed.

Figure 2, the transplanted senescent cells should also be GFP positive to track their disappearance (see 1).

---

## [Author Response]

*1) The authors use the concept that since there are no EdU positive cells that are also SA-βgal positive, then the senescent cells must be cleared by macrophages. However, the investigators fail to mention or consider that if there was a SA-βgal positive cell that incorporated EdU then the SA-βgal staining would go away. This needs to be addressed very carefully, since this is a central point in the manuscript*.

*Related to this, it is not clear from the text if the transplanted senescent cells are also expressing a fluorescent marker; it seems not. If not, it would be better to have experiments tracking the senescent cells with an independent marker such as RFP. As it stands, although unlikely, cells could be reversing the senescence and losing B-gal expression rather than being cleared from the blastema*.

We acknowledge that there are two possible explanations for the disappearance of SAbgal senescent cells: their active clearance, or the reversal of the senescent state. These two possibilities can be distinguished with the use of retrovirally labelled nGFP+ senescent cells. Therefore, we have incorporated a full set of experiments, tracing the clearance of senescent cells genetically labeled with nGFP. As seen in Figure 3, the rate of clearance of nGFP+ senescent cells corresponds to the rate of clearance of unlabelled senescent cells, providing conclusive evidence that senescent cells are indeed cleared from salamander tissues. Please see Figure 3. We have incorporated a sentence covering this point in the subsection headed “Efficient surveillance of senescent cells in adult salamander tissues”.

*2) The authors should consider depleting macrophages from salamanders using the technique they described*, *then induce regeneration and provide quantitative data on βgal positive cells with time in a real animal. Do the numbers of senescent cells still disappear in macrophage depleted animals similarly to salamanders with macrophages?*

We welcome the reviewers’ suggestion, which we have indeed considered. At present, it is not possible to carry out the experiment suggested by the reviewers, as macrophages are essential for regeneration and the blastema will not develop upon macrophage depletion (22).

What happens in vitro when salamander A1 senescent cells are mixed with macrophages?

This is an interesting suggestion. However, we cannot currently address this due to a lack of methods to study the macrophage-senescent cell interaction in the salamander context (we have tested senescent cells-total blood cultures, however the blood elements are highly reactive; we are now trying to successfully sort and culture macrophages in order to study this critical interaction).

*3) Many immune processes are dependent on macrophages and there are several different cells types that could have been examined. The lack of molecular mechanisms of the pathways from senescent cells to them being cleared is a serious weakness*.

We strongly agree with the reviewers that elucidating the molecular mechanisms underlying the clearance of senescent cells is paramount. We have set out to address this from an early start, but faced the difficulties of working with a system that is not as well developed as human or mice. Indeed, the characterization of the immune system of the salamander is still in its infancy, with only a couple of reports published so far, and a serious lack of reagents to study it. The present study is concerned to establish that the surveillance process operates in regenerating and normal salamander tissues, but, if reagent developments allow, further studies will focus on the mechanisms that underlie this important process.

*That both normal and senescent cells are cleared but with a different time course suggests the mechanisms of clearance may be more complex than the investigators claim*.

We do not agree with this last statement. As shown in Figure 3, all cells suffer from an initial reduction in numbers (likely due to a combination of the implantation method, tissue integration and accessibility to nutrients) however normal cells clearly persist while senescent cells are actively removed (with a half-life of 9 days).

*4) It is disappointing that the investigators did not attempt to identify what is secreted by the senescent cells? Is it the same SASP factors as demonstrated in mammals? This would have added some mechanistic information to the study*.

*Also, nothing is detailed about the mechanism of how senescent cells can induce senescence in normal cells*.

We fully agree with the reviewers that adding a characterization of the SASPs would enrich the study, as it may shed light into the way senescent cells affect their own clearance and their environment, as shown by the pioneering work of the Campisi lab and others. Therefore, we have incorporated this analysis to Figure 1, based on a combination of human cytokine and protease arrays (although with the limitation that many non-conserved factors could not be evaluated as they result in negative reads). We show that senescent salamander cells secrete many factors, which have been reported to be secreted by mammalian senescent cells in various contexts (as SASPs differ between different species, cell types and modes of senescence). Note additional Methods and Results.

Regarding the mechanism of paracrine senescence, we cannot speculate at this point which factors are responsible for such effect, as this requires an extensive analysis. It is possible that TGF-β, which has been shown to mediate paracrine senescence in a mammalian context (as shown by the Gil Lab, [1]), is implicated. Certainly, our experiments suggest that a secreted factor is involved. However, this is an issue that merits further investigation.

*5) The role of paracrine-senescence is poorly investigated and does not contribute significantly to the paper. The fact that it can be induced is clear, as has been reported in a number of other papers (Acosta et al., Nature Cell Biol, 2013 reference is missing). And this could help to explain how the co-injection of senescent cells with normal cells favors the removal of the normal. However, it could as easily be that the senescent cells simply recruit immune cells to the site of injection, which then remove any exogenous cells.*
Figure 4
*suggests that paracrine senescence may contribute, but this should be quantified to show this is the mechanism*.

We do not share the reviewers’ view on our dataset on the paracrine effect, as our data clearly demonstrates, by various means, that salamander cells are able to induce senescence in normal cells in a paracrine fashion (using similar assays to those commonly used within the field). This constitutes a valuable insight regarding the conservation of this aspect of cellular senescence in salamanders (particularly as this is the first thorough characterisation of senescence in the amphibian branch of the animal kingdom). Furthermore, as recommended by the reviewer, we have incorporated a quantification of the % SAbgal+/nGFP+ cells after co-implantation of control (nGFP+) and senescent (nGFP-) cells. The results suggest that, indeed, paracrine senescence does contribute to the clearance of this mixed population (See Figure 4), providing an explanation for the data shown in Figure 3.

We thank the reviewers for highlighting the missing reference, which has now been incorporated.

*Furthermore, it is claimed that senescent cells are found in clusters in the spleen because of a paracrine-senescence effect. It is more likely that senescent cells injected in one site are being removed to the spleen, and this is why they appear in* “*clusters*”*. Such claims should be experimentally validated*.

We agree that this is an issue that merits further exploration. Hence, we have removed this piece of data and will explore the issues raised by the reviewers further.

*6) The use of UV as an irreversible senescence inducer is not robust. UV treatment of mammalian cells (depending on the dose) results in cell repair or death. UV induces p53 and a variety of different types of DNA strand breaks, and depending on the dose, many cells can repair and survive. Almost nothing is mentioned about a dose curve for these experiments. If the investigators use a dose that induces senescence but is reversible (low doses) do the cells clear similarly to normal or senescent cells*.

We thank the reviewers for this suggestion. We have now incorporated an additional condition for implantation: nGFP cells treated with a low UV dose (1J/m2) which leads to <15% senescence induction (which results in persistence of nGFP cells within the limb, much like nGFP control cells). Please see Figure 3 and Figure 3—figure supplement 1.

*7) It appears that senescence can only be induced in salamander cells by the stabilization of p53, but this is completely omitted from the Discussion. This is confusing, given that the same group has recently shown that down regulation of p53 correlates with the timing of senescence induction (Yun et al., PNAS 2013). In addition, nutlin treatment impairs limb regeneration if treated at the blastema stage, but increased it at the regeneration phase*.

*Given the importance of p53 in the regulation of senescence and their previous paper, this needs more detailed explanation*.

During regeneration, p53 is indeed downregulated during blastema formation (excluding the first 48h), however its activity comes down 3-fold from high levels in normal tissues (In contrast, the levels of p53 activity are very low in A1 cells). Furthermore, p53 is still functional during regeneration (we show in our PNAS paper that complete inhibition of p53 impairs regeneration, suggesting that p53 activity, albeit lower than in normal libs, is present during the critical stages that coincide with senescent induction). Finally, it is important to note that the measurements of p53 activity in regeneration represent the average of p53 activity in the entire structure, while in the case of senescent cells one is only looking at a minority population (<10%). Hence, the conditions during regeneration are not inconsistent with an environment permissive for p53- induced senescence.

*8) The fact that there is an increase in senescence with wounding is well established in other studies (Jun and Lau, Nature Cell Biology, July 2010; Krizhanovsky et al., Cell, 2008), as is the fact that senescent cells are cleared by macrophage-mediated removal (Xue et al., Nature, 2007; Krizhanovsky et al., Cell, 2008; Kang et al., Nature 2011). These papers should be cited*.

We thank the reviewers for drawing this to our attention. We have incorporated the additional missing references (Kang et al. was already present in the original manuscript). In addition, we have incorporated an additional, very recent reference to senescence in wound healing (18).

9) Most figures need better explanation in the text, describing exactly what was done, and in enough detail to follow. The figure legends similarly need more detail.

We have made several improvements in both figure legends and text in order to improve the overall clarity of the manuscript.

*Many figures and experiments also lack quantification and controls. While the result can be taken at face-value as a description of what was observed, more detail would make the data much more convincing*.

*E.g.*
Figure 1
*needs control staining of non-senescent cells to show these markers of senescence are increased under these conditions.*

We have incorporated the respective control staining of non-senescent cells (Figure 1). In addition, we have added a quantification of the percentage of cells DHR123 (ROS) positive (Figure 1).

Figure 2*, it is quite difficult to see the senescent cells and where they are located. Higher resolution images, with some reference to localization/distribution are needed*.

We thank the reviewers for their suggestion. We have incorporated a new Figure supplement with high power images of senescent cells in different areas of the regenerating limb (Figure 2—figure supplement 1).

Figure 2*, the transplanted senescent cells should also be GFP positive to track their disappearance (see 1)*.

We have incorporated this data (Figure 3).